# Navigational health literacy and health service use among higher education students in Alentejo, Portugal - A cross-sectional study

Jorge Rosário[1,2,3]*, Sara Simões Dias[3,4,5], Sónia Dias[6], Ana Rita Pedro[7]

**1** Polytechnic Institute of Beja, Beja, Portugal, **2** Institute for Research and Advanced Training, University of Évora, Évora, Portugal, **3** Comprehensive Health Research Centre, CHRC, University of Évora, Évora, Portugal, **4** citechcare - Center for Innovative Care and Health Technology, Polytechnic of Leiria, Leiria, Portugal, **5** School of Health Sciences, Polytechnic of Leiria, Campus 2 - Morro do Lena, Alto do Vieiro, Leiria, Portugal, **6** NOVA National School of Public Health, Public Health Research Centre, Comprehensive Health Research Center, CHRC, REAL, CCAL, NOVA University Lisbon, Lisbon, Portugal, **7** NOVA National School of Public Health, Public Health Research Centre, Comprehensive Health Research Center, CHRC, REAL, CCAL, NOVA University Lisbon, Lisbon, Portugal

☯ These authors contributed equally to this work.

\* Jorge.olhoazul@ipbeja.pt

## Abstract

### Introduction

The navigational health literacy of higher education students is fundamental to effective health management and successful health navigation, thereby improving health outcomes and overall well-being. Assessing the general and navigational health literacy levels of these students is crucial for developing targeted interventions and facilitating informed decision-making on health-related issues. This study aimed to identify the levels of general and navigational health literacy, characterise access to and utilisation of healthcare services, and analyse the differences between the mean general and navigational health literacy indices and determinants among higher education students in the Alentejo region of southern Portugal.

### Methodology

A descriptive and cross-sectional study was conducted between 25 May and 12 September 2023 with 1979 higher education students. An online structured questionnaire comprising the Portuguese version of the European Health Literacy Survey Questionnaire – 16 items (*HLS-EU-PT-Q16*) and the Navigational Health Literacy Scale (*HLS$_{19}$-NAV*), both from the European Consortium, was used. Sociodemographic data, presence of chronic disease, perceived health status, perceived availability of money for expenses, and healthcare access and utilisation variables were included. The study data were analysed using independent samples t-test, one-way ANOVA, and post hoc Bonferroni test, followed by multiple linear regression analyses at a

**Data availability statement:** All relevant data are within the manuscript.

**Funding:** This research was funded by Fundação para a Ciência e Tecnologia (FCT, Portugal) through national funds to the Associated Laboratory in Translation and Innovation Towards Global Health REAL (LA/P/0117/2020). No additional external funding was received for this study. The funder had no role in study design, data collection and analysis, decision to publish, or preparation of the manuscript.

**Competing interests:** The authors have declared that no competing interests exist.

significance level of 0.05. Multiple linear regression analysis was performed to identify factors associated with both general and navigational health literacy. The study protocol was approved by the ethics committee of the University of Évora, and all participants provided written informed consent.

## Results

Most students (86.8%) exhibited limited general health literacy, while 13.2% demonstrated adequate health literacy. Inadequate navigational health literacy was observed in 73.4% of students. Students with lower mean general and navigational health literacy were more likely to have utilised health services. Students with chronic conditions, recent use of urgent or emergency services, and difficulties in accessing healthcare had lower health literacy. Conversely, those enrolled in health-related courses, those with good financial resources and those who had not used health services during their course had higher health literacy. In addition, lower navigational health literacy was found among displaced students, those with chronic conditions and those who had recently consulted a doctor. Higher navigational health literacy was associated with enrolment in health-related courses and adequate general health literacy.

## Conclusion

The findings highlight the significant influence of demographic and academic factors on general and navigational health literacy among higher education students. The prevalence of limited general and navigational health literacy underscores a significant challenge for students, institutions, and health policy makers. Effective health literacy interventions should take these factors into account. Future research should examine longitudinal changes in health literacy and evaluate the impact of targeted educational programmes.

---

## Introduction

Health literacy (HL) encompasses the knowledge, motivation, and competencies to access, understand, appraise, and apply health information in order to make judgements and take decisions in everyday life about healthcare, disease prevention, and health promotion [1]. The European Health Literacy Survey (HLS-EU) project emphasises health literacy as a modifiable determinant of health, integrating public health and individual approaches [2–6]. It is a part of current health, social and educational policies [7], and depends on individual competence as well as environmental factors, resources and context [1].

The Integrated Model of Health Literacy, which is widely used in Europe, provides a comprehensive framework for understanding how health literacy influences health behaviours and outcomes. It identifies key determinants and factors that impact health literacy levels (antecedents) and their resulting health outcomes

(consequences) [5]. This model outlines four core competencies of health literacy: access (seeking and obtaining health information), understand (comprehending the information), appraise (interpreting, evaluating, and filtering the information), and apply (using the information to make informed health decisions). These competencies are relevant across the health continuum, encompassing healthcare, disease prevention, and health promotion [5]. Their effectiveness is influenced by organisational structures and the availability of resources, which affect individuals' ability to make informed health decisions for their own wellbeing and that of those around them [8].

Navigational health literacy (NAV-HL), a subset of health literacy, is relevant for effective navigation of the complex, fragmented health care system. It encompasses an individual's knowledge, motivation, and skills to access, understand, appraise, and apply information needed to navigate health care systems and services efficiently [9–12].

Navigational health literacy can be understood at three levels: the system level (macro), which deals with the organisation and functioning of the health care system; the organisational level (meso), where users organise information for decision making; and the interactional level (micro), which emphasises active engagement in the processing of health information [9].

Effective navigation of health services involves expressing preferences and gathering information from health professionals in order to participate in decision making and future health planning [9,12]. The complexity of health systems leads to challenges in accessing care, as reported by people with multiple health needs who experience disjointed and unco-ordinated services [13]. Navigating the system is crucial given its different sectors and services [11,14]. Examining both general and navigational health literacy is crucial for understanding how individuals access and use health services.

Research consistently links lower health literacy with higher healthcare utilisation and costs, poorer health outcomes, and greater financial burdens [15–19]. Inadequate health literacy often results in increased medical errors and communication difficulties with providers [20–22], highlighting the critical need for improved health literacy initiatives.

Although limited health literacy has been demonstrated among Portuguese university students [23–28], the specific levels of general and navigational health literacy among students in the Alentejo region remain unexplored. This gap highlights the importance of focusing on this region to address the specific challenges related to health literacy.

The Alentejo region in southern Portugal, which covers 30% of the country's territory, faces unique challenges such as low tertiary education rates, high drop-out rates, low population density, and risks of depopulation, making it crucial to promote health literacy and healthy behaviours, especially in rural areas. Assessing general health literacy, navigational health literacy, and access to and utilisation of health services among higher education students in the Alentejo region is essential for policy makers, academic institutions, health services, and health professionals. This assessment is fundamental for the development of targeted interventions to improve health literacy, reduce vulnerability and improve health outcomes among higher education students in the Alentejo region. To the best of our knowledge, the lack of previous studies on access, utilisation of health services, general and navigational health literacy among higher education students in the Alentejo may compromise health planning measures, health policies and the design of health literacy interventions promoted by health professionals.

In this context, our study aims to address the following research questions: (i) what are the levels of general and navigational health literacy among higher education students in the Alentejo region of southern Portugal?; (ii) what are the characteristics of access to and use of health services among these students?; and (iii) how do levels of general and navigational health literacy relate to determinants such as socio-demographic variables, presence of chronic diseases, perceived health status, perceived availability of money for expenses, and access to and use of health services?

To answer these questions, the study will: (i) identify levels of general and navigational health literacy; (ii) characterise access to and use of health services; and (iii) analyse the differences between levels of health literacy, and determinants such as socio-demographic variables, chronic diseases, perceived health status, perceived availability of money for expenses, and access to and utilisation of health services among higher education students in the Alentejo region.

 

This study will contribute to a deeper understanding of health literacy, and its differences in access to and use of health services among higher education students in the Alentejo. This knowledge will guide the development of targeted interventions to improve health outcomes and empower higher education students to make informed health decisions, ultimately facilitating better access to and use of health services.

## Materials and methods

### Study design and setting

An observational, descriptive, quantitative, cross-sectional study was conducted using an online survey between May 25 and September 12, 2023, involving four unique public higher education institutions in the Alentejo region (statistical territorial unit - NUT II) of Portugal: Polytechnic Institute of Beja, Polytechnic Institute of Portalegre, Polytechnic Institute of Santarém, and University of Évora. Following the acquisition of institutional approval, all undergraduate and integrated master's students received an email invitation via the educational institutions' communication channels. This email contained detailed information regarding the study, as well as a link to access the questionnaire. The invitation provided a comprehensive overview of the study's objectives, the participation conditions, and the significance of participation. The initial section of the questionnaire encompassed all the aforementioned elements. Upon completion and understanding of the information, the students provided their consent to participate. To enhance the response rate and ensure a representative sample, follow-up reminders were sent to students, and course coordinators gave alerts during meetings.

### Study population and sample size

The study population comprised all students, from all academic years, enrolled in undergraduate and integrated master's degree programs totalling 13135 students in the 2022–2023 academic year [29] who voluntarily consented to participate and understood written Portuguese. Postgraduate students were excluded because the study aimed to design health interventions or policies for undergraduate students.

Based on similar studies [23,30–32], the minimum sample size was determined using standard statistical procedures [33,34], taking into account a 95% confidence level, a population proportion of 50% (to maximise variance and ensure a conservative estimate) and a 3% margin of error. After applying the finite population correction, the required sample size was adjusted to 952 students. To account for an expected non-response rate of 20%, the final target sample size was increased to 1143 students. However, the final dataset comprised 1979 participants, exceeding the required minimum. The augmentation in sample size is attributable to a response rate that surpassed expectations, propelled by the accessibility of the online survey, institutional support, and student interest in the subject matter. This enlarged sample size fortified the statistical power, enhanced subgroup representativeness and facilitated more detailed analyses, thus augmenting the robustness and precision of the results [34].

### Study participants and data collection

The Portuguese versions of the European Health Literacy Survey instruments (developed by the European Consortium World Health Organization Action Network on Measuring Population and Organizational Health Literacy – M-POHL), specifically the 16-item *HLS-EU-PT-Q16* for health literacy and the 12-item $HLS_{19}$-*NAV* for navigational health literacy, were used [9,35–37]. The questionnaire also covered health services access, use, and socio-demographic characteristics. It was distributed via an online platform with a unique hyperlink in the study invitation. Each participant could complete the questionnaire only once to ensure data integrity.

Responses were collected electronically using software that securely recorded and stored data in accordance with ethical research standards, ensuring the anonymity and confidentiality of participants. Regular reminders were sent to maximise participation. To prevent non-responses, the questionnaire was designed to be mobile-friendly, ensuring accessibility

on various devices, including smartphones and tablets, and was concise to minimise completion time. Friendly reminders were sent to reduce non-response bias.

**Institutional approval and promotion.** Approval was obtained from each institution's governing body via email, which detailed the study's aims and requested endorsement. After approval, institutions promoted the questionnaire link and consent form to students through social media and email lists. Course coordinators emphasized the survey's importance, and members of the National Academic Health Literacy Network assisted with dissemination and awareness efforts. This multi-channel approach aimed to increase visibility and response rates, with clear instructions to complete the questionnaire only once.

**Addressing potential biases.** The potential biases associated with online data collection were recognised. All students had free access to the internet and the questionnaire was designed to work on a variety of electronic devices. To mitigate these biases, the questionnaire was made available on a variety of devices commonly used by students, including those available in higher education institutions, and technical support was provided where necessary. The multi-channel promotion strategy aimed to reach a wide range of student populations, thereby improving diversity and ensuring a representative sample. Efforts were made to emphasise the importance of responses and to maintain anonymity to reduce response bias.

## Questionnaire/Measurements

The questionnaire started with informed consent and age verification. Participants then completed the *HLS-EU-PT-Q16*, followed by questions on health services access and use, the *HLS$_{19}$-NAV*, and socio-demographic questions. The 50-item questionnaire was pre-tested with a subset of 31 participants.

## Characterisation of the instrument HLS-EU-PT-Q16

The Portuguese version of the 16-item *HLS-EU-PT-Q16*, developed by the European Health Literacy Consortium, was used to assess health literacy. The HLS19 instrument, developed by M-POHL for the International Health Literacy Population Survey 2019–2021, evaluated the difficulty/ease of accessing, comprehending, appraising, and applying information in various domains [37–39].

Each item on the scale has four valid possible responses (very easy, easy, difficult, and very difficult) and also the option "don't know/refusal" (it was considered missing). Each response corresponds to a numerical value: 0 – don't know, 1 – very difficult, 2 – difficult, 3 – easy, 4 – very easy, and 5 – don't know/refusal. Only responses with at least 80% of items rated 1–4 were included in the scoring; others were excluded.

The average score for all items was calculated and converted into an index according to the European Health Literacy Consortium guidelines. The calculation of the index score was performed according to the following formula: (mean-1)*(50/3), where mean corresponds to the average of items on the scale, 1 – the minimal value of the mean, 3 – the range of the mean, and 50 – the chosen maximum value of the new index scores [2,40–42].

Categories were created based on the cut-offs: excellent (>42–50); sufficient (>33–42); problematic (>25–33); and inadequate (0–25) [2,41,42]. Subsequently, the categories were dichotomised into limited (combining inadequate and problematic health literacy categories) and adequate health literacy (combining sufficient and excellent health literacy categories).

The Cronbach's alpha coefficient for *HLS-EU-PT-Q16*, which indicates internal consistency, was .89 overall, and .783 for the health care subdomain, .724 for the disease prevention subdomain, and .703 for the health promotion subdomain [37,38].

## Characterisation of the instrument HLS$_{19}$-NAV

Navigational health literacy, a specific aspect of health literacy concerning navigation skills, was assessed using the *HLS$_{19}$-NAV* instrument [9,12,43]. It is dedicated to evaluating navigational health literacy within the framework of the European Health Literacy Survey (HLS$_{19}$-EU). The instrument comprises 12 items, with responses on a Likert scale,

scored with 1 = "very difficult"; 2 = "difficult"; 3 = "easy"; 4 = "very easy"; and 5 = "don't know/refusal", which was considered as missing. The instrument evaluates how easily participants can access, understand, appraise, and apply information about navigating the healthcare system. Only responses with at least 80% of items rated 1–4 were included in the scoring, otherwise they were excluded. For analysis purposes, responses were dichotomized: responses indicating ease were categorised as "very easy" and "easy", while responses indicating difficulty were categorised as "difficult" and "very difficult". Responses categorised as "very easy" and "easy" were aggregated and converted to a unified metric on a scale of 0–100. The $HLS_{19}$-NAV index represents the percentage of items answered with "easy" and "very easy", with higher scores indicating higher navigational health literacy [43]. For descriptive analysis, scores were categorised into four levels: excellent (>83.3), sufficient (>66.6), problematic (>50), and inadequate (≤50) [12]. The $HLS_{19}$-NAV instrument's validation in Portugal and its Cronbach's α of.939 highlight its reliability [39].

### Independent variables

Sociodemographic variables included age, gender, relocation to attend the course, cohabitation with a healthcare professional, and the highest educational level of parents or the person with whom the student lived, based on the International Standard Classification of Education (ISCED) [44].

Academic background included academic institution, academic year, being a finalist, completion of a previous course in the health field, and frequenting a health-related course.

Additional variables included the presence of chronic disease, self-perceived health status (categorised as unsatisfactory or satisfactory) [31], perceived financial availability (categorised as poor or good) [39].

Variables considered for healthcare services access and utilisation included: (i) use of urgent or emergency services in the last 24 months; (ii) medical consultation in the past 12 months; (iii) hospitalisation for more than one day within the last 12 months; (iv) outpatient hospital visits in the last 12 months (e.g., day hospital or ambulatory surgery - in and out on the same day); (v) absence from work/school due to health problems in the last 12 months; (vi) utilisation of health services while attending the course; (vii) main reason for last use of health services while attending the course; (viii) difficulty accessing a scheduled appointment; (ix) difficulty accessing an urgent appointment; (x) difficulty in using health services; (xi) main reason hindering the use of health services; and (xii) first action to take if feeling ill in non-urgent situations where there is no reason to call 112 (emergency number in Europe).

### Data analyses

The statistical analysis was performed using IBM-SPSS version 29. Descriptive analysis assessed the distribution of independent variables among participants. The assumption of population normality and homogeneity of variances, prerequisites for statistical tests, were supported by the Central Limit Theorem. Given normal distribution and homogeneity of variances, parametric tests (Student's t -test and ANOVA) were employed for group comparisons. Post hoc Bonferroni tests were conducted following a significant result in the overall ANOVA to identify specific group differences. Multiple linear regression analysis was performed to identify factors associated with both general and navigational health literacy. A statistical significance level of $p < 0.05$ was adopted. Results will be presented in aggregate form, with additional analyses to explore variations by institution and academic year if necessary.

### Ethical considerations

The study was approved by the Ethics Committee of the University of Évora (Document number 22091) and the Scientific Council. Students provided their informed consent digitally. All students were informed of the aims of the study through a written statement of voluntary and informed consent on the first page of the questionnaire. Students who did not consent were not given access to the questionnaire. Confidentiality, anonymity, and de-identification of data were strictly

maintained throughout the study. To ensure confidentiality, data were stored on secure servers with access restricted only to researchers involved in data analysis, using secure passwords as a requisite for access.

## Results

### Students' characteristics

The sample comprised 1979 higher education students. The mean age of the students was 21.42 ± 2.95 years. The majority of students identified as female, 57.8% (n = 986), while 42.2% (n = 720) identified as male. A total of 76.8% (n = 1519) of the students were displaced from their usual place of residence, 95.6% (n = 1892) were not living with a health professional, and 38.8% (n = 768) had completed upper secondary education (ISCED 3). Only 9.6% (n = 190) had previously completed a course in health care.

Most of the students were in their third year (n = 732; 37.0%), with the smallest proportion in their fifth year (n = 60; 3.0%). The majority of respondents (n = 1340; 67.7%) were not final year students and a significant proportion (n = 1789; 90.4%) had not completed a health-related course prior to the survey. In addition, 72.0% (n = 1425) of respondents reported that they were not currently enrolled in a health-related course.

In terms of health status, 63.8% (n = 1262) of students did not have a chronic disease, while 79.8% (n = 1574) reported an unsatisfactory health status, and in relation of the perceived availability of financial resources for expenses, 96.2% (n = 1904) of students indicated that they perceived poor availability of money for them. Table 1 summarises the sample, the mean differences of the general and navigational health literacy between student characteristics.

### Healthcare services access and utilisation

Table 2 shows the mean differences in the general health literacy index and the navigational health literacy index, taking into consideration the access to and utilisation of healthcare services. The results show that a significant proportion of students used urgent or emergency services in the last 24 months (n = 1421; 71.8%) and sought medical consultation in the last 12 months (n = 1166; 58.9%). Hospitalisation for more than one day was less frequent (n = 144; 7.3%), while 18.8% (n = 373) of students reported outpatient visits.

A significant proportion of students (n = 1428; 72.2%) reported missing work or school due to health problems, while a substantial number (n = 1499; 75.7%) used health services during their studies. The main reasons for seeking healthcare were unexpected health problems (n = 452; 30.2%) and problems exacerbated by delayed treatment (n = 527; 26.6%).

The study also highlighted challenges such as difficulty accessing scheduled appointments (n = 1602; 80.9%) and identified the physical distance to the service (n = 1188; 60.0%). Interestingly, in non-urgent situations where emergency services were not required, the majority of students (n = 1188; 60.0%) chose to go to the emergency department of a public hospital.

### General health literacy

Upon analysis of the mean and standard deviation of individual responses on the *HLS-EU-PT-Q16*, it was found that the highest mean score (2.3 ± 1.0) was observed for the item assessing the difficulty students experienced in understanding instructions from their doctor or pharmacist regarding prescribed medicine. Conversely, the lowest mean scores (1.9 ± .8) were observed for the items related to determining when to seek a second opinion from another doctor, finding information on managing mental health issues, and assessing the reliability of health risk information in the media (see Table 3).

Table 4 shows the distribution of the health literacy index according to the competencies of health literacy: access, understand, appraisal, and apply. The total overall general health literacy index ranged from 0 to 50. The mean overall health literacy index was 18.6 ± 11.7, indicating inadequate health literacy (with scores ranging from a minimum of 0 to a maximum of 47.9). Within specific health literacy domains, scores of 18.9 ± 13.8 were observed for health care, 18.4 ± 12.8 for disease prevention and 18.3 ± 11.4 for health promotion.

**Table 1. Mean differences of the general and navigational health literacy between student characteristics (N = 1979).**

| Characteristics of the higher education students - independent variables | | N (%) | HL Mean Index (SD) [0;50] | p | NAV HL Mean index (SD) [0;100] | p |
|---|---|---|---|---|---|---|
| Age (grouped into classes) | [16;20] | 803 (40.6) | 17.7 (11.4) | .001 Ω | 23.8 (28.0) | <.001 Ω |
| | [21;25] | 911 (46.0) | 17.8 (11.0) | | 25.6 (30.8) | |
| | >=26 | 265 (13.4) | 24.2 (13.0) | | 44.6 (36.3) | |
| Gender (N = 1706) | Feminine | 986 (57.8) | 21.8 (11.1) | <.001† | 33.6 (31.1) | <.001† |
| | Masculine | 720 (42.2) | 17.5 (11.9) | | 27.4 (32.7) | |
| Displaced from usual residence to attend the course | Yes | 1519 (76.8) | 17.4 (11.2) | <.001† | 24.3 (30.0) | <.001† |
| | No | 460 (23.2) | 22.5 (12.2) | | 37.7 (33.1) | |
| Cohabitation with a health professional | Yes | 87 (4.4) | 26.1 (7.8) | <.001† | 36.8 (29.4) | .188† |
| | No | 1892 (95.6) | 18.3 (11.7) | | 27.0 (31.3) | |
| Level of education of the student's parents or the person with whom the student lived | No formal education/basic (ISCED 0–2) | 611 (30.9) | 18.8 (11.8) | 0.517 Ω | 28.5 (31.2) | <.001 Ω |
| | Upper secondary (ISCED 3) | 768 (38.8) | 18.2 (11.4) | | 24.2 (29.1) | |
| | Higher education (ISCED 5–8) | 600 (30.3) | 18.9 (11.8) | | 30.4 (33.6) | |
| Academic year | First | 275 (13.9) | 27.1 (8.7) | <.001 Ω | 37.8 (25.5) | <.001 Ω |
| | Second | 501 (25.3) | 17.1 (11.1) | | 19.5 (25.7) | |
| | Third | 732 (37.0) | 14.7 (10.3) | | 18.8 (28.9) | |
| | Fourth | 411 (20.8) | 21.8 (12.8) | | 46.3 (34.3) | |
| | Fifth | 60 (3.0) | 18.2 (9.3) | | 20.8 (33.9) | |
| Finalist | Yes | 639 (32.3) | 20.5 (12.2) | <.001† | 32.3 (34.1) | <.001† |
| | No | 1340 (67.7) | 17.7 (11.3) | | 25.1 (29.5) | |
| Previous completion of a course in healthcare | Yes | 190 (9.6) | 25.6 (9.9) | <.001† | 32.9 (26.5) | <.001† |
| | No | 1789 (90.4) | 17.9 (11.6) | | 26.8 (31.7) | |
| Healthcare-related course | Yes | 554 (28.0) | 28.8 (7.9) | <.001† | 50.1 (28.1) | <.001† |
| | No | 1425 (72.0) | 14.6 (10.4) | | 18.6 (27.6) | |
| Chronic disease | Yes | 717 (36.2) | 15.0 (11.3) | <.001† | 18.9 (29.3) | <.001† |
| | No | 1262 (63.8) | 20.6 (11.4) | | 32.3 (31.3) | |
| Perceived health status (n = 1973) | Satisfactory | 399 (20.2) | 30.0 (7.2) | <.001† | 43.7 (25.2) | <.001† |
| | Unsatisfactory | 1574 (79.8) | 15.5 (10.8) | | 23.4 (31.3) | |
| Perceived availability of money for expenses | Good | 75 (3.8) | 34.8 (6.7) | <.001† | 50.3 (23.9) | <.001† |
| | Bad | 1904 (96.2) | 18.0 (11.3) | | 26.5 (31.2) | |

*Legend: Ω – ANOVA; † - t Student*

Fig 1 shows the percentage distribution of health literacy levels across general health literacy and domains. The general health literacy categories show that 86.8% of students had limited health literacy (67.5% inadequate and 19.3% problematic), while 13.2% had adequate health literacy (11.4% sufficient and 1.8% excellent). According to the dimensions of health literacy, limited health literacy of 83.0% was found for health promotion, 80.9% for disease prevention and 79.3% for health care.

## Navigational health literacy

When analysing the distribution of responses from the $HLS_{19}$-NAV instrument, as shown in Table 5, the items with the highest mean scores were assessing a health service needed for a health problem (2.1 ± .8), deciding for a particular health service (2.1 ± .8), and understanding how to book appointments at a particular health service (2.1 ± .8).

The mean navigational health literacy index was 29.1±31.6, with a minimum value of 0 and a maximum value of 100. The navigational health literacy categories of the students were as follows Fig 2. The results indicated that 75.6% of the students demonstrated inadequate navigational health literacy.

**Differences among students' characteristics, healthcare services utilisation, and mean indexes of general and navigational health literacy**

**Students' characteristics.** Table 1 shows the significant differences between the characteristics of higher education students and the mean of the General Health Literacy (GHL) and Navigational Health Literacy (NAV-HL) indexes. Health literacy indexes varied significantly by age group, with students aged ≥26 years (n=265; 13.4%) showing higher GHL (24.2±13.0) and NAV-HL (44.6±36.3) compared to younger students (16–20 years) ($p < .001$). The Bonferroni post hoc analysis reveals that individuals aged 26 and above exhibit significantly higher scores on both the GHL index and NAV-HL index compared to those aged 16–20 and 21–25 ($p < .001$ for both indices). Specifically, the GHL index shows no significant difference between the 16–20 and 21–25 age groups ($p = 1.0$). For the NAV-HL index, mean differences are -20.7 ($p < .001$) between 26 and 16–20, and -18.9 ($p < .001$) between 26 and 21–25, with no significant difference between the 16–20 and 21–25 age groups (mean difference: -1.7, $p = .686$).

Female students (n=986; 57.8%) had significantly higher scores in both general (21.8±11.1) and navigational health literacy (33.6±31.1) mean index compared to male students (n=720, 42.2%), with 17.5±11.9 for general and 27.4±32.7 for navigational health literacy mean index.

Students who were displaced to attend their course (n=1519; 76.8%) demonstrated lower general and navigational health literacy mean index (GHL: 17.4±11.2; NAV-HL: 24.3±30.0) compared to those who were not displaced (n=460; 23.2%) (GHL: 22.5±12.2; NAV-HL: 37.7±33.1) ($p < .001$).

Students living with health professionals (n=87; 4.4%) had notably higher general health literacy scores (26.1±7.8) compared to those without such cohabitants (18.3±11.7), for navigational health literacy it was not observed significant statistical differences. It was not observed statistically significant differences between the level of education of the student's parents or the person with whom the student lived at general health literacy, but for navigational health literacy it was observed that students with parents with higher education level, has higher navigational health literacy mean index ($p < .001$).

First-year students (n=275; 13.9%) had significantly higher health literacy scores (GHL: 27.1±8.7; NAV HL: 37.8±25.5) compared to students in subsequent years. Fourth-year students (n=411; 20.8%) showed an increase in general health literacy (21.8±12.8). The Bonferroni post hoc analysis reveals that first-year students have significantly higher GHL scores compared to students in the second through fifth years, with notable mean differences of 10.0, 12.4, 6.3, and 8.9, respectively ($p < .001$), while significant declines in scores are observed in second, third, fourth, and fifth-year students compared to first-year students, with no significant differences between these latter years; additionally, NAV-HL also peaks in the first year but significantly decreases in later years, although fourth-year students show an increase compared to some other years.

Students enrolled in healthcare-related courses (n=554; 28.0%) demonstrated higher general and navigational mean levels (GHL: 28.8±7.9; NAV-HL: 50.1±28.1) compared to their peers in non-health-related courses (n=1425; 72.0%). Students with chronic diseases (n=717; 36.2%) reported lower health literacy scores (GHL: 15.0±11.3; NAV-HL:18.9±29.3) than those without chronic conditions (n=1262; 63.8%). Students with satisfactory perceived health status (n=399; 20.2%) had significantly higher health literacy scores (GHL: 30.0±7.2; NAV-HL: 43.7±25.2). Similarly, those perceiving good availability for expenses (n=75; 3.8%) had higher scores (GHL: 34.8±6.7; NAV-HL: 50.3±23.9).

**Health services access and utilisation.** Students who used urgent or emergency services in the previous 24 months (n=1421;71.8%) had a significantly lower GHL mean index (17.9±11.1) and NAV-HL mean index (24.3±29.3) compared to those who did not (n=558; 28.2%), with an HL mean index of 23.3±10.3 and NAV-HL mean index of 35.4±33.7

**Table 2. Means differences of the general and the navigational health literacy index taking into consideration the access to and utilisation of healthcare services (N = 1979).**

| Healthcare services access and utilisation | | N (%) | HL Mean Index [0;50] | p | NAV-HL Mean Index [0;100] | p |
|---|---|---|---|---|---|---|
| Use of urgent or emergency services in the last 24 months | Yes | 1421 (71.8) | 17.9 (11.1) | <.001† | 24.3 (29.6) | <.001† |
| | No | 558 (28.2) | 23.3 (10.3) | | 35.4 (33.7) | |
| A medical consultation within the last 12 months. | Yes | 813 (41.1) | 18.9 (11.5) | .419† | 31.7 (31.0) | .186† |
| | No | 1166 (58.9) | 18.8 (11.5) | | 24.4 (31.1) | |
| A period of hospitalisation of more than one day within the past 12 months (hospital as an inpatient) | Yes | 144 (7.3) | 14.4 (21.1) | <.001† | 26.6 (34.0) | .135† |
| | No | 1835 (92.7) | 19.3 (11.0) | | 27.5 (31.0) | |
| Attended a hospital on an outpatient basis in the last 12 months (e.g., Day Hospital or Ambulatory Surgery - in and out the same day - hospital as a day patient) | Yes | 373 (18.8) | 17.0 (11.6) | .003† | 29.5 (32.6) | .211† |
| | No | 1606 (81.2) | 19.4 (11.0) | | 26.9 (30.9) | |
| Absence from work/school due to health problems in the last 12 months | Yes | 1428 (72.2) | 18.0 (11.1) | <.001† | 26.0 (31.7) | .090† |
| | No | 551 (27.8) | 21.2 (11.2) | | 31.0 (29.8) | |
| Utilisation of health services while attending the course | Yes | 1499 (75.7) | 18.1 (11.0) | <.001† | 26.4 (32.2) | <.001† |
| | No | 480 (24.3) | 21.1 (11.3) | | 30.7 (28.1) | |
| Main reason for last use of health services while attending the course Reason (n = 1499) | Related to chronic illness | 207 (8.9) | 17.9 (13.8) | <.001Ω | 20.8 (29.2) | <.001Ω |
| | Unexpected problem | 452 (30.2) | 21.0 (10.7) | | 29.9 (33.0) | |
| | Problem made worse by not completing treatment | 109 (9.2) | 16.9 (10.8) | | 29.5 (33.3) | |
| | Aggravation of a problem for which you had not sought help | 527 (26.6) | 13.8 (9.7) | | 19.6 (29.7) | |
| | Other reason | 313 (20.9) | 27.2 (8.9) | | 47.3 (31.2) | |
| Difficulty accessing a scheduled appointment (n = 1940) | Very difficult+Difficult | 1602 (80.9) | 16.3 (10.6) | <.001† | 22.3 (30.1) | .040† |
| | Easy+Very easy | 338 (17.1) | 29.2 (6.4) | | 51.1 (26.5) | |
| Difficulty accessing an urgent appointment (n = 1924) | Very difficult+Difficult | 1762 (89.0) | 17.7 (10.8) | <.001† | 24.9 (30.7) | .093† |
| | Easy+Very easy | 162 (8.2) | 28.4 (8.9) | | 53.0 (27.8) | |
| Difficulty in using health services | Yes | 1637 (82.7) | 17.2 (10.5) | <.001† | 23.2 (30.1) | <.001† |
| | No | 342 (17.3) | 26.5 (10.8) | | 47.6 (28.6) | |

*(Continued)*

**Table 2.** (Continued)

| Healthcare services access and utilisation | | N (%) | HL Mean Index [0;50] | p | NAV-HL Mean Index [0;100] | p |
|---|---|---|---|---|---|---|
| **Main reason that hinders the use of health services** | Financial reasons | 265 (13.4) | 17.8 (11.4) | **<.001**$^{\Omega}$ | 43.1 (28.4) | **<.001**$^{\Omega}$ |
| | Reason related to the physical distance to the service | 1188 (60.0) | 15.9 (9.1) | | 20.3 (29.8) | |
| | Reason related to the delay in being attended to | 82 (4.1) | 24.9 (8.3) | | 26.5 (28.3) | |
| | Unfamiliarity with the available system | 207 (10.5) | 13.5 (9.5) | | 39.4 (32.1) | |
| | Other reason(s) | 237 (12.0) | 24.5 (10.1) | | 35.1 (30.3) | |
| **The first thing to do if you feel ill in non-urgent situations where there is no reason to call 112** | Goes to a health centre for a consultation without an appointment | 265 (13.4) | 27.5 (7.7) | **<.001**$^{\Omega}$ | 43.1 (28.4) | **<.001**$^{\Omega}$ |
| | Goes to the emergency department of a public hospital | 1188 (60.0) | 15.0 (10.1) | | 20.3 (29.8) | |
| | Makes an appointment at the health centre | 82 (4.1) | 26.5 (12.6) | | 26.5 (28.3) | |
| | Calls SNS24 | 207 (10.5) | 21.1 (11.4) | | 39.4 (32.1) | |
| | Other procedure | 237 (12.0) | 24.2 (9.5) | | 35.1 (30.3) | |

Legend: NAV-HL - Navigational Health Literacy; HL - Health Literacy; $^{\Omega}$ – ANOVA; $^{\dagger}$ - t Student

($p < .001$ for both). In contrast, there was no significant difference in the GHL and NAV-HL indices between students who had a medical consultation within the last 12 months (n = 813; 41.1%) and those who did not (n = 1166; 58.9%), with p-values of.419 for GHL and.186 for NAV-HL.

Students hospitalized for more than one day in the previous 12 months (n = 144; 7.3%) had a significantly lower GHL mean index (14.4 ± 21.1) compared to those who were not hospitalized (n = 1835; 92.7%), who had an GHL mean index of 19.3 ± 11.0 ($p < .001$). However, the NAV-HL indexes did not differ significantly ($p = .135$). Those who attended a hospital on an outpatient basis in the last 12 months (18.8%, n = 373) had significantly lower GHL mean indexes (17.0 ± 11.6) compared to those who did not (n = 1606; 81.2%), with an GHL mean index of 19.4 ± 11.0 ($p = .003$). The NAV-HL index differences were not significant ($p = .211$).

Students who were absent from work or school due to health problems in the previous 12 months (n = 1428; 72.2%) had significantly lower GHL mean indices (18.0 ± 11.1) compared to those who were not absent (27.8%, n = 551), who had an HL mean index of 21.2 ± 11.2 ($p < .001$). The difference in NAV-HL indices was not significant ($p = .090$).

Relatively of the health services utilisation during course attendance students (n = 1499; 75.7%) had significantly lower GHL (18.1 ± 11.0) and NAV-HL (26.4 ± 32.2) indices compared to non-users (n = 480; 24.3%), who had HL of 21.1 ± 11.3 and NAV-HL of 30.7 ± 28.1 ($p < .001$ for both).

Significant differences in HL and NAV-HL indices were observed based on the reason for the last use of health services ($p < .001$ for both). For example, those that had an aggravation of a problem for which they had not sought help (n = 527; 26.6%,) had lower indices (GHL: 13.8 ± 9.7, NAV-HL: 19.6 ± 29.7) compared to those with other reasons (GHL: 27.5 ± 8.9, NAV-HL: 47.3 ± 31.2). The Bonferroni post hoc analysis reveals that students seeking health services for chronic diseases have lower GHL and NAV-HL indices compared to those with unexpected problems or other reasons,

**Table 3. Distribution of observed frequencies, percentages, mean and standard deviation of responses to each item on the HLS-EU-PT-Q16 scale (N = 1979).**

| Area | On a Scale from Very Easy to Very Difficult, How Easy Would You Say It Is To… | 1- Very difficult (N, %) | 2 - Difficult (N, %) | 3 - Easy (N, %) | 4 - Very easy (N, %) | Mean±SD [1;4] | 5 - Don't know/ Refusal (N, %) [missing] |
|---|---|---|---|---|---|---|---|
| HC | 1. find information on treatments of illness that concern you | 734; 37.1 | 472; 23.9 | 521; 26.3 | 231; 11.7 | 2.1±1.0 | 21; 1.0 |
| HC | 2. find out where to get professional help when you are ill? | 657; 33.2 | 704; 35.6 | 464; 23.4 | 147; 7.4 | 2.0±.9 | 7;.4 |
| HC | 3. understand what your doctor says to you? | 643; 32.5 | 580; 29.3 | 568; 28.7 | 183; 9.2 | 2.1±.9 | 5;.3 |
| HC | 4. understand your doctor's or pharmacist's instruction on how to take a prescribed medicine | 638; 32.2 | 515; 26.0 | 505; 25.5 | 320; 16.2 | 2.2±1.0 | 1;.1 |
| HC | 5. judge when you may need to get a second opinion from another doctor? | 625; 31.6 | 830; 41.9 | 425; 21.5 | 73; 3.7 | 1.9±.8 | 26; 1.3 |
| HC | 6. use information the doctor gives you to make decisions about your illness? | 564; 28.5 | 769; 38.8 | 506; 25.6 | 103; 5.2 | 2.0±.8 | 37; 1.9 |
| HC | 7. follow instructions from your doctor or pharmacist? | 548; 27.7 | 573; 29.0 | 584; 29.5 | 274; 13.8 | 2.3±1.0 | 0;.0 |
| DP | 8. find information on how to manage mental health problems like stress or depression? | 602; 30.4 | 879; 44.4 | 384; 19.4 | 89; 4.5 | 1.9±.8 | 25; 1.3 |
| DP | 9. understand health warnings about behaviour such as smoking, low physical activity and drinking too much? | 616; 31.1 | 552; 27.9 | 469; 23.7 | 333; 16.8 | 2.2±1.0 | 9;.5 |
| DP | 10. understand why you need health screenings? | 586; 29.6 | 609; 30.8 | 468; 23.6 | 296; 15.0 | 2.2±1.0 | 20; 1.0 |
| DP | 11. judge if the information on health risks in the media is reliable? | 661; 33.4 | 801; 40.5 | 410; 20.7 | 95; 4.8 | 1.9±.8 | 12;.6 |
| DP | 12. decide how you can protect yourself from illness based on information in the media? | 571; 28.9 | 822; 41.5 | 489; 24.7 | 89; 4.5 | 2.0±.8 | 8;.4 |
| HP | 13. find out about activities that are good for your mental well-being? | 582; 29.4 | 689; 34.8 | 526; 26.6 | 170; 8.6 | 2.1±.9 | 12;.6 |
| HP | 14. understand advice on health from family members or friends? | 705; 35.6 | 589; 29.8 | 574; 29.0 | 101; 5.1 | 2.0±.9 | 10;.5 |
| HP | 15. understand information in the media on how to get healthier? | 732; 37.0 | 569; 28.7 | 554; 28.0 | 124; 6.3 | 2.0±.9 | 0;.0 |
| HP | 16. judge which everyday behaviour is related to your health? | 617; 31.2 | 559; 28.2 | 602; 30.4 | 195; 9.9 | 2.1±.9 | 6;.3 |

Legend: HC – Healthcare, DP – Disease Prevention; HP – Health Promotion; SD – Standard Deviation.

while those with other reasons have the highest indices; significant differences are noted among all groups. Students who reported difficulty in accessing scheduled appointments (n = 1602; 80.9%) had significantly lower GHL (16.3 ± 10.6) and NAV-HL (22.3 ± 30.1) indices compared to those who found it easy/very easy (17.1%, n = 338), who had GHL of 29.2 ± 6.4 and NAV-HL of 51.1 ± 26.5 ($p < .001$ for GHL; $p = .040$ for NAV-HL). Similar trends were observed for urgent appointments, where those who reported difficulties (n = 1762; 89.0%) had a significantly lower GHL (17.7 ± 10.8) compared to those who did not (n = 162; 8.2%), who had a GHL of 28.4 ± 8.9 ($p < .001$). The difference in NAV-HL was not significant ($p = .093$).

Students who experienced difficulties in using health services (n = 1637; 82.7%) had significantly lower GHL (17.2 ± 10.5) and NAV-HL (23.2 ± 30.1) indices than those who did not (n = 342; 17.3%), who had GHL of 26.5 ± 10.8 and NAV-HL of 47.6 ± 28.6 ($p < .001$ for both).

Significant differences in GHL and NAV-HL were observed according to reasons for not seeking health care ($p < .001$ for both). Unfamiliarity with the available system was associated with lower GHL (13.5 ± 9.5) and the reason related to the physical distance to the service for NAV-HL (26.5 ± 28.3) ($p < .001$).

**Table 4. Distribution of the health literacy index mean and standard deviation according to the competencies of health literacy: access, understand, appraisal, and apply.**

| Health literacy dimensions | Health literacy competencies and health literacy index [0;50] | | | | |
|---|---|---|---|---|---|
| | All | Find/access | Understand | Appraise | Apply |
| Health care | 18.9 ± 13.8 | 18.1 ± 15.2 | 20.0 ± 16.5 | 16.2 ± 13.8 | 19.9 ± 14.8 |
| Disease Prevention | 18.4 ± 12.8 | 16.3 ± 13.8 | 20.6 ± 17.3 | 16.1 ± 14.3 | 17.4 ± 14.1 |
| Health Promotion | 18.3 ± 11.4 | 19.0 ± 15.7 | 17.2 ± 12.3 | 19.8 ± 16.4 | Not available, no data existing |
| General Health Literacy Index | 18.6 ± 11.7 | 18.0 ± 13.0 | 19.4 ± 13.2 | 17.4 ± 10.6 | 19.0 ± 13.6 |

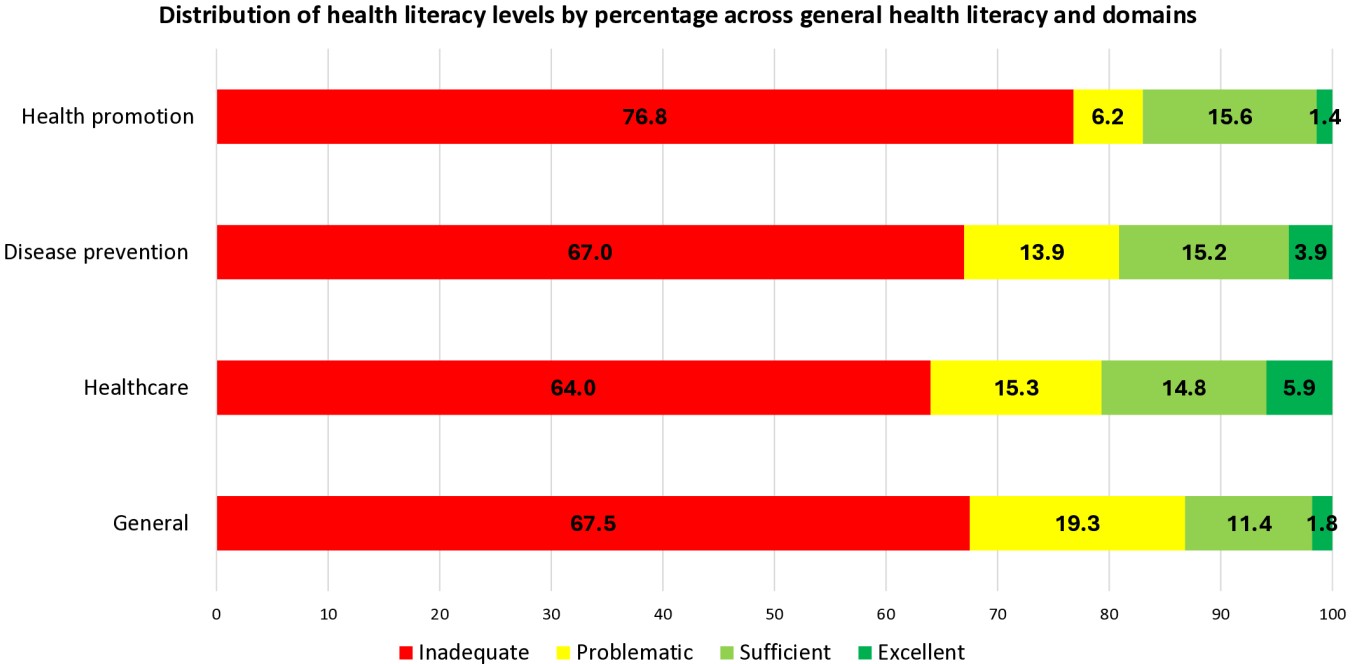

**Fig 1. Distribution of health literacy levels by percentage across general health literacy domains.**

The first action taken in non-urgent health situations also showed significant differences ($p < .001$ for both GHL and NAV-HL). For example, those who went to a health centre without an appointment (n = 265; 13.4%) had higher GHL (27.5 ± 7.7) and NAV-HL (43.1 ± 28.4) than those who went to the emergency department (n = 1188; 60.0%), who had GHL of 15.0 ± 10.1 and NAV-HL of 20.3 ± 29.8.

The descriptive statistics show significant differences in the GHL and NAV-HL indices for different methods of accessing health services, with the highest GHL indices observed for those who consulted pharmacists and private clinics, and the lowest for those who used public hospital emergency services, highlighting the differences in health literacy according to the method of accessing health services.

## The relationship between the independent variables and general health literacy using a linear regression model

The linear regression model for general health literacy, shown in Table 6, identifies several significant predictors, ranked in order of importance. The strongest positive predictor is perceived availability of money for expenses, which has a

**Table 5.** Distribution of observed frequencies, percentages, mean and standard deviation of responses to each item on the Navigational Health Literacy scale (N = 1979).

| On a Scale from Very Easy to Very Difficult, How Easy Would You Say It Is to... | 1 - Very difficult (N; %) | 2- Difficult (N; %) | 3 - Easy (N; %) | 4 - Very easy (N; %) | Mean ± SD [1;4] | 5 - Don't know/ Refusal (N; %) |
|---|---|---|---|---|---|---|
| 1.understand information on how the health care system works [e.g., which type of health services are available] | 776; 39.2 | 601; 30.4 | 481; 24.2 | 118; 6.0 | 1.9±.9 | 3;.2 |
| 2.judge which type of health service you need in case of a health problem | 560; 28.3 | 750; 37.9 | 545; 27.5 | 116; 5.9 | 2.1±.8 | 8;.4 |
| 3.judge to what extent your health insurance covers a particular health service [e.g., are there any co-payments] | 542; 27.4 | 897; 45.4 | 404; 20.4 | 94; 4.7 | 2.0±.8 | 42; 2.1 |
| 4.understand information on ongoing health care reforms that might affect your health care | 546; 27.6 | 978; 49.4 | 340; 17.2 | 77; 3.9 | 1.9±.7 | 38; 1.9 |
| 5.find out about your rights as a patient or user of the health care system | 519; 26.2 | 885; 44.7 | 444; 22.4 | 114; 5.8 | 2.0±.8 | 17;.9 |
| 6.decide for a particular health service [e.g., choose from different hospitals] | 548; 27.7 | 739; 37.3 | 528; 26.7 | 162; 8.2 | 2.1±.9 | 2;.1 |
| 7.find information on the quality of a particular health service | 601; 30.4 | 848; 42.8 | 384; 19.4 | 131; 6.6 | 2.0±.8 | 15;.8 |
| 8.judge if a particular health service will meet your expectations and wishes on health care | 607; 30.6 | 858; 43.4 | 437; 22.1 | 65; 3.3 | 1.9±.8 | 12;.6 |
| 9.understand how to get an appointment with a particular health service | 564; 28.5 | 756; 38.2 | 545; 27.5 | 113; 5.7 | 2.1±.8 | 1;.1 |
| 10.find out about support options that may help you to orientate yourself in the health care system | 586; 29.6 | 847; 42.8 | 459; 23.2 | 70; 3.5 | 2.0±.8 | 17;.9 |
| 11.locate the right contact person for your concern within a health care institution [e.g., in a hospital] | 642; 32.4 | 857; 43.3 | 361; 18.2 | 100; 5.1 | 1.9±.8 | 19; 1.0 |
| 12.stand up for yourself if your health care does not meet your needs | 742; 37.5 | 778; 39.3 | 294; 14.9 | 139; 7.0 | 1.9±.8 | 26; 1.3 |

Legend: SD – Standard Deviation; NAV-HL – Navigational Health Literacy.

### Distribution of navigational health literacy levels by percentage across navigational health literacy

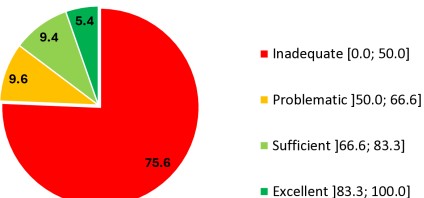

- Inadequate [0.0; 50.0]
- Problematic ]50.0; 66.6]
- Sufficient ]66.6; 83.3]
- Excellent ]83.3; 100.0]

**Fig 2.** Distribution of navigational health literacy levels by percentage across navigational health literacy.

significant positive effect on general health literacy (B = 6.4, $p < .001$). This highlights the crucial role of financial stability in improving individuals' health literacy.

Experiencing ease of access to a scheduled appointment (categorised as easy or very easy) also shows a significant positive association with general health literacy (B = 5.4, $p < .001$) highlighting the importance of accessible health services.

Satisfactory perceived health status is another significant positive predictor, strongly associated with higher levels of general health literacy (B = 5.6, *p* < .001), suggesting that individuals who perceive their health positively are more likely to have better health literacy.

Age is positively associated with general health literacy (B = 2.7, *p* < .001), suggesting that older students tend to have higher health literacy.

Conversely, the presence of a chronic condition is associated with a significant negative impact on overall health literacy (B = -2.1, *p* < .001), highlighting the challenges people with chronic conditions face in maintaining high levels of health literacy.

### The relationship between navigational health literacy and independent variables using a linear regression model

The linear regression model for navigational health literacy, detailed in Table 7, identifies several significant predictors. Use of urgent or emergency services in the last 24 months is associated with a significant negative effect on navigational health literacy (B = -8.5, p < .001). In addition, difficulty using health services shows a strong negative association with navigational health literacy (B = -14.5, *p* < .001). Conversely, taking a health-related course is positively associated with navigational health literacy (B = 4.1, *p* = .003). However, not having completed a health-related course is significantly negatively associated with navigational health literacy (B = -6.7, *p* < .001).

In addition, higher levels of general health literacy, categorised as sufficient or excellent compared to inadequate or problematic, are strongly positively associated with navigational health literacy (B = 32.3, *p* < .001).

## Discussion

### Summary of findings

This cross-sectional study aimed to identify the levels of general and navigational health literacy, characterise the access to and utilisation of health services, and analyse the differences in mean general and navigational health literacy indices across socio-demographic variables, presence of chronic disease, perceived health status, perceived availability of money for expenses, and access to and utilisation of health services among higher education students in the Alentejo region of southern Portugal.

The findings provide insights into the health literacy landscape, particularly concerning navigational health literacy among higher education students. To our knowledge, there have been no prior studies on access, utilisation of healthcare services, general and navigational health literacy among higher education students in Alentejo, making this study pioneering.

The study encompasses 1979 higher education students, for all the public academic institutions in Alentejo (NUTS II). In terms of age and gender, the results obtained are similar to the national distribution of higher education students in 2022/2023 [29,45,46]. On a scale 0–50 the general health literacy (GHL) mean index among higher education students in Alentejo was 18.6 ± 11.7. According to the competencies for health literacy it was observed that the lowest score at appraise health information (17.4 ± 10.6) and the highest at understand health information (19.4 ± 13.2). Concerning to health literacy dimensions the lower general health literacy was observed in the health promotion dimension (18.3 ± 11.14) and the highest at healthcare (18.9 ± 13.8). These findings are similar to those of Pedro et al. (2022), who highlighted that approximately 44% of higher education students in Portugal have limited health literacy [23]. Furthermore, they are consistent with studies conducted among Portuguese higher education students [24–26,28,47,48] and evidence for the general population [37,39,49–52]. This study highlights the significant proportion of students who may face challenges in understanding and processing health-related information, which is essential for making informed health decisions and managing their health effectively. The implications of this finding are profound, as inadequate health literacy can lead to poorer health outcomes, increased hospitalisations and higher healthcare costs due to poor management of health conditions.

**Table 6. Linear regression model – General Health Literacy.**

| Variables | B | Standard Error (SE) | CI 95% | | p value |
|---|---|---|---|---|---|
| | | | Lower | Higher | |
| Intercept | -6.1 | 1.2 | -8.6 | -3.6 | <.001 |
| Chronic disease (Yes) | -2.1 | .4 | -3.0 | -1.2 | <.001 |
| Perceived availability of money for expenses (good) | 6.4 | 1.1 | 4.1 | 8.6 | <.001 |
| Difficulty accessing a scheduled appointment (easy/very easy) | 5.4 | .02 | 4.9 | 6.0 | <.001 |
| Age | 2.7 | .3 | 2.1 | 3.3 | <.001 |
| Perceived health status (satisfactory) | 5.6 | .6 | 4.3 | 6.9 | <.001 |

Legend: SE – Standard Error; CI – Confidence Interval

In another study, Amaral et al. (2021) investigated health literacy levels among residents of Viseu, a town in the interior of Portugal, and found similarly low levels of health literacy. This study showed that low levels of health literacy can affect an individual's ability to make informed health decisions [25]. Specifically, the research showed that limited health literacy was associated with difficulties in navigating the healthcare system, understanding medical instructions and adhering to prescribed treatments. The findings from Amaral et al. are consistent with broader trends observed in other regions, suggesting a more general problem that goes beyond isolated communities [27,50,53].

The higher prevalence of health literacy observed in our study, compared with the findings of Pedro et al. (44%) and Amaral et al. (32%), and more in line with Santos et al. (66%), can be attributed to several factors [25,27,37]. The socio-demographic and educational characteristics of the Alentejo (NUTS II) student population may play a role in shaping health literacy levels. In addition, increased exposure to different sources of health information, as highlighted by Santos et al, may have contributed to the observed differences in health literacy [28]. These findings underscore the dynamic nature of health literacy, influenced by education, access to reliable health information, and engagement with health-related content, and support the need for targeted health literacy interventions in the future [27].

Nutbeam (2000), Baker (2006), and Sørensen et al. (2012) criticised traditional models of health literacy for neglecting social and contextual factors [5,41,54,55]. Nutbeam proposed a broader view, distinguishing between functional, interactive and critical literacy, and emphasising the need to critically evaluate and apply health information. Baker pointed out that traditional measures fail to address social and economic inequalities, while Sørensen et al. (2012) highlighted the role of social interactions and community context [5,41,54,55]. Pelikan (2022) supports these views and advocates a holistic, contextual approach to health literacy, emphasising the importance of individual, social and societal factors [56].

The current study builds on these findings by focusing on higher education students in the Alentejo region of southern Portugal. This population is particularly important as it represents a group transitioning into adulthood, where health literacy becomes increasingly important for maintaining health and preventing disease. The results show that higher education students also had low levels of general and navigational health literacy, echoing the concerns raised by Pedro et al. and Amaral et al. [23,25].

Navigational health literacy (NAV-HL) was the most challenging, with 85.2% of the respondents in our sample having inadequate and problematic levels and a mean navigational health literacy index of 29.1±31.6 on a scale of 0–100. These findings are consistent with some previous research in this area and highlight the need for future development of tailored interventions to address this issue [39].

The observed association between navigational health literacy levels and overall health literacy underlines their interdependence. Improving the environment for higher education students, who are young adults in transition, is critical given their vulnerability to current and future health challenges. For interventions to improve the health literacy and navigational health literacy of higher education students, it's relevant in this study dimensions of health promotion and disease

**Table 7. Linear regression model – Navigational Health Literacy.**

| Variables | B | Standard Error (SE) | CI 95% | | p value |
|---|---|---|---|---|---|
| | | | Lower | Higher | |
| Intercept | 11.0 | 3.2 | -1.7 | 11.1 | <.001 |
| Use of urgent or emergency services in the last 24 months (yes) | -8.5 | 1.4 | -11.3 | -5.7 | <.001 |
| Difficulty in using health services (yes) | -14.5 | 1.7 | -18.2 | -11.5 | <.001 |
| Healthcare-related course (yes) | 4.1 | 1.4 | 1.4 | 6.9 | .003 |
| Previous completion of a course in healthcare (no) | -6.7 | 1.2 | -9.2 | -4.2 | <.001 |
| Sufficient/Excellent compared to inadequate/problematic health literacy level | 32.3 | 1.9 | 28.5 | 36.0 | <.001 |

Legend: SE – Standard Error; CI – Confidence Interval

prevention. The competencies that can be improved are finding and appraising health information. In the specific context of the Alentejo, with socio-demographic characteristics as elderly/aging, it is important to empower the young adults to make health decisions for their health and well-being.

The study reveals significant differences in GHL and NAV-HL among higher education students based on their patterns of healthcare use. Students who had used urgent or emergency services in the previous 24 months had significantly lower GHL and NAV-HL indices than those who had not, highlighting a critical gap in health literacy among those who frequently use emergency care. Similarly, students who had been hospitalised or made outpatient visits also had lower GHL indices, although NAV-HL did not show a significant difference in these cases. In addition, students who missed school or work due to health problems had lower GHL indices, further suggesting that frequent and complex interactions with the health care system correlate with lower health literacy.

A plausible reason for these lower levels of health literacy among higher education students could be their transitional life stage, which often involves increased independence and responsibility, but limited prior exposure to complex health care systems. This transitional period may lead to inadequate health literacy due to a lack of formal health education and support systems as students navigate new environments and manage their own health needs [48,57,58]. The challenge is compounded by the rapid changes in their social and academic lives, which may reduce their ability to engage effectively with health information and services [54]. These findings highlight the importance of targeted health literacy interventions tailored to the specific needs of students during this transitional period.

The analysis highlights the influence of barriers to accessing and navigating health services on health literacy levels. Students who reported difficulties in making appointments and using health services had significantly lower GHL and NAV-HL indices. In particular, financial constraints were associated with poorer health literacy than other barriers, such as cultural factors [59–61]. he data also showed that the mode of access to health care had an impact on health literacy, with higher indices for those who used health centres or private clinics compared to public hospital emergency departments. These findings highlight the need for targeted interventions to address literacy inequalities and improve access to healthcare, particularly for students facing systemic barriers [5,54,61,62]. Higher education students often have lower health literacy due to life transitions and systemic barriers, such as financial constraints, which affect their ability to access and use health services effectively. This is compounded by difficulties in navigating complex health systems and accessing timely care, which further exacerbates health literacy inequalities [1,5,23,54,57,58,61,63].

**General and navigational health literacy levels among characteristics of the students**

The mean age of the students was 21.4±2.9 years, and students aged 26 years or older had a higher mean general health literacy index and a higher mean navigational health literacy index compared to younger age groups ($p<.001$). This suggests that maturity and potentially more life experience contribute positively to health literacy [54]. Most students

were female (57.8%) and had higher mean general (21.8±11.1) and navigational (33.6±31.1) health literacy index scores compared to males ($p < .001$).

Students who were displaced from their usual residence to attend the course had lower mean general health literacy index scores (17.4±11.2) and lower mean navigational health literacy index scores (24.3±30.0), ($p < .001$). The challenges associated with displacement, such as adjusting to a new environment, may affect their health literacy. Living with health professionals resulted in higher mean general health literacy index (26.1±7.8) and mean navigational health literacy index (36.8±29.4) scores ($p < .001$). Exposure to health-related discussions and behaviours at home is likely to improve understanding and navigation of health information [64].

Although there were no significant differences in mean general health literacy index scores based on parental education level, there was a trend towards higher mean navigational health literacy index scores among those with parents with higher education. This suggests that a more educated home environment may better prepare students with skills to navigate health systems [54].

First-year students had higher mean general health literacy (27.1±8.7) and fourth-year students had higher mean navigational health literacy (46.3±34.3) scores ($p < .001$). Students enrolled in health-related courses had significantly higher health literacy (28.8±7.9) and navigational health literacy (50.1±28.1) scores ($p < .001$). This highlights the impact of health education on improving health literacy [61,65].

Students with chronic diseases had lower general health literacy (15.0±11.3) and navigational health literacy (18.9±29.3) scores ($p < .001$). Chronic health conditions can be overwhelming for students, potentially leading to lower levels of health literacy.

Students who perceived their health status as satisfactory had higher general health literacy (30.0±7.2) and navigational health literacy (43.7±25.2) scores ($p < .001$). Positive health perceptions may correlate with proactive health behaviours and better health literacy.

Students with a positive perception of their availability of money for expenses reported higher health literacy (34.8±6.7) and navigational health literacy (50.3±23.9) scores ($p < .001$). Financial stability is likely to facilitate access to health resources and education, thereby improving health literacy. Further research is needed to explore additional determinants of health literacy, and their implications for public health interventions. The findings highlight the importance of different factors associated with health literacy levels [10].

### General and navigational health literacy levels among students' health services access and utilisation

The results of this study reveal significant differences in health literacy (HL) and navigational health literacy (NAV-HL) among students, highlighting critical barriers to accessing and using health services effectively.

Health service users had significantly lower HL and NAV-HL indices than non-users. This inverse relationship, with frequent service users having lower levels of literacy, may reflect the compounding difficulties faced by those with lower health literacy, including a reduced ability to search for, understand, and use health information effectively. This is consistent with the work of Sørensen et al. (2012), who highlighted that low health literacy is both a cause and a consequence of poor health outcomes, leading to a cycle of increased service use and poorer health management [5,61].

The significant differences in HL and NAV-HL based on the reasons for the last health service use provide critical insights. Students with unresolved health problems or those who did not seek help initially reported significantly lower levels of literacy. This suggests a reactive rather than proactive approach to health management, where students may only seek help when problems worsen. Such findings are consistent with research by Berkman et al. (2011), which highlighted that individuals with lower health literacy often delay seeking care until conditions worsen, resulting in higher health care utilisation and costs [57].

Accessibility issues are another major concern. Students who experienced difficulties in scheduling and attending appointments had significantly lower GHL and NAV-HL. This suggests that logistical barriers, such as scheduling conflicts

and long waiting times, have a direct impact on students' ability to manage their health effectively. Nutbeam (2008) has also pointed out that navigating the health system requires both individual skills and a supportive health system infrastructure, which appears to be lacking in this student population [66].

The finding that students who had difficulty accessing urgent care also had lower HL underlines the acute challenges of managing sudden health needs. This is consistent with Weiss et al. (2004) who documented that lower health literacy is associated with higher emergency department visits, often due to a lack of understanding of when and how to seek appropriate care [67].

Overall, these findings highlight the urgent need to address both individual and systemic factors that influence health literacy and service navigation. Interventions need to focus on educational strategies to improve students' health literacy while streamlining access to health services. The integration of user-friendly health information systems, as advocated by Baker (2006), could significantly improve students' ability to navigate and use health services effectively, ultimately improving health outcomes in this vulnerable population [55].

## Strengths and limitations

This study makes a significant contribution to the understanding of health literacy and health services use among higher education students and has several key strengths. First, it represents a pioneering effort as the first comprehensive assessment of health literacy levels and health service use patterns in the Alentejo region of southern Portugal. This novel research fills a critical gap due to the lack of previous baseline data, and provides a foundation for future research.

Methodologically, the study demonstrates a robust design by using validated instruments, including the Portuguese version of the European Health Literacy Survey Questionnaire (*HLS-EU-PT-Q16*) and the Navigational Health Literacy Scale (*HLS$_{19}$-NAV*). The *HLS-EU-PT-Q16* has been psychometrically validated for the Portuguese population, demonstrating strong internal consistency (Cronbach's alpha = 0.89), test-retest reliability and construct validity in assessing health literacy levels [38]. Similarly, the *HLS19-NAV* has undergone cross-national validation in several European contexts, including Portugal, confirming its reliability (Cronbach's alpha > 0.80) and construct validity for measuring navigational health literacy [39,68]. These rigorous validation processes ensure methodological robustness and strengthen the accuracy and reliability of data collection and analysis [2,37,39,41,38,68].

The study benefits from a diverse sample of 1979 students from different institutions, which increases the generalisability of the findings. Although there are limitations in capturing variability across all higher education institutions, the diversity of the sample provides valuable insights into the health literacy levels of this population. Comprehensive data analysis techniques, including independent samples t-tests, one-way ANOVA, post hoc Bonferroni tests, and multiple linear regression, were used to thoroughly explore the relationship between health literacy levels and various determinants. According to Setia (2016), this rigorous analytical approach strengthens the validity and reliability of the study's conclusions, in line with the recommended standards [60].

The practical implications of these findings are significant, identifying key demographic and academic factors that influence health literacy and providing a basis for the development of targeted educational programmes. This is consistent with Nutbeam's (2000) advocacy for structured health education programmes to improve health literacy [54] and with Berkman et al. (2011) support for policy development aimed at reducing health inequalities and improving access to health services [57]. The study is relevant to health professionals, academics, community organisations and policy makers, and can inform their strategic health decisions.

However, it is important to consider the limitations of the study. Although the sample is diverse, it may not be fully representative of all higher education institutions, which may affect the generalisability of the findings. The cross-sectional design limits the ability to draw causal inferences. In addition, the study did not examine the impact of institutional health programmes on health literacy, nor did it address psychological and behavioural factors, such as academic stress and lifestyle, which may significantly influence health literacy [69]. A potential limitation of this study was the inclusion of health sciences students alongside students from other disciplines, which may have influenced the data analysis.

Another limitation is the reliance on self-reported data, which may introduce social desirability bias. Students may have given answers that they perceived to be more socially acceptable, potentially biasing the results. This bias is a common feature of studies using self-report measures, as highlighted by researchers such as Paulhus (1991), who emphasised the influence of social desirability on survey responses [59].

Several strategies have been developed to minimise these biases, such as regular reminders about how to complete the questionnaire (in different formats and at different times), availability to clarify doubts, technical support to resolve difficulties, and reassurance at all times about the anonymity and confidentiality of responses.

## Implications for future research and interventions

These findings have important implications for policy planning, programme development, and future research directions. It is essential that further studies deepen the understanding of the relationship between general and navigational health literacy and health inequalities. The observed association between general and navigational health literacy and the independent variables highlights their interrelationship, which has been highlighted by Sørensen et al. (2012), who argue that improving health literacy can lead to a reduction in health inequalities [5].

Improving the environment for higher education students, who are young adults in transition, is critical given their vulnerability to current and future health challenges. The study highlights that access to, and use of health services is more prevalent among students with lower general and navigational health literacy. This finding is consistent with research by Berkman et al. (2011), which suggests that lower health literacy is associated with increased healthcare utilisation due to poor health management [57].

These findings highlight the importance of investing in targeted health literacy interventions to improve access to and use of health services, with a particular focus on navigational skills. Workshops, mini-courses or lectures on health issues and service use may be examples of interventions that could be designed with particular attention to health care and health promotion. Tailored interventions that improve students' ability to navigate health services could reduce some of the barriers identified in this study. As Nutbeam (2000) notes, effective health literacy programmes are essential to empower individuals to make informed health decisions [54].

Future research should use longitudinal designs to monitor changes in health literacy over time. It should also evaluate specific interventions aimed at improving health literacy levels. Such studies would provide a deeper understanding of the development of health literacy and its long-term effects on health outcomes, as suggested by Coulter and Ellins (2007) [62]. It is recommended that future studies explicitly differentiate between health sciences students and those in other fields in order to facilitate a more precise investigation of any disparities in health literacy that may be evident between these two groups. Furthermore, future research should seek to include a more diverse pool of students and include additional variables such as socioeconomic status, health behaviours, and psychological factors that may influence health literacy. By addressing these variables, researchers can gain a more nuanced understanding of the determinants of health literacy and its impact on health service use.

Our study identified unanticipated health problems and worsening conditions as the main drivers of emergency service use among students. This highlights the urgent need for further research into the relationship between lifestyle factors and healthcare seeking behaviour. Specifically, future research should assess the extent to which avoidable emergency visits are associated with modifiable lifestyle factors, explore the role of health literacy in promoting engagement in preventive healthcare, and develop targeted interventions to reduce overuse of emergency services among students.

Given that 1,574 students (79.5%) reported their health as unsatisfactory, it is important to further investigate the underlying factors contributing to these perceptions. Future research should disaggregate self-reported health into physical, mental and social dimensions, identify key determinants influencing these perceptions, and explore effective intervention strategies to improve student well-being.

## Conclusion

The study shows that levels of general and navigational health literacy among higher education students in Alentejo, Portugal, are notably lower than anticipated. This finding highlights a significant challenge, given the crucial role health literacy plays in accessing and using health services effectively. The study identifies several predictors of health literacy among students, including the presence of chronic conditions, frequency of health-related coursework, perceived financial resources, use of urgent or emergency services, and difficulties in accessing health services.

It has been demonstrated that students with limited health literacy encounter obstacles that have the potential to exert a detrimental influence on their well-being and health outcomes. These results reinforce the established associations between health literacy and healthcare utilisation, thus underlining the need for structured health education to improve health literacy and mitigate health inequalities.

The findings highlight the urgent need to develop and implement tailored health literacy interventions that address the specific needs of higher education students. These interventions should focus on health promotion and disease prevention, and aim to improve students' ability to make informed health decisions and navigate health services effectively. This is particularly important given the influential role of students in shaping wider societal health decisions.

Future research should build on these findings by using longitudinal designs to track changes in health literacy over time and by examining additional variables such as socioeconomic status, health behaviours and psychological factors. Such research will provide a more comprehensive understanding of the determinants of health literacy and inform more effective intervention strategies.

Addressing the challenges identified is critical to reducing systemic barriers and ensuring equitable access to health care. By improving the health literacy and navigational skills of university students, we can promote better health outcomes and contribute to a more informed and health-conscious society.

## Author contributions

**Conceptualization:** Jorge Rosário, Sara Simões Dias, Sónia Dias, Ana Rita Pedro.

**Formal analysis:** Jorge Rosário, Sara Simões Dias, Sónia Dias, Ana Rita Pedro.

**Investigation:** Jorge Rosário, Sara Simões Dias, Sónia Dias, Ana Rita Pedro.

**Methodology:** Jorge Rosário, Sara Simões Dias, Sónia Dias, Ana Rita Pedro.

**Writing – original draft:** Jorge Rosário, Sara Simões Dias, Sónia Dias, Ana Rita Pedro.

**Writing – review & editing:** Jorge Rosário, Sara Simões Dias, Sónia Dias, Ana Rita Pedro.

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
