## [Decision Letter · Decision Letter 0]

12 Feb 2025

PONE-D-24-32494Navigational Health Literacy and Health Service Use Among Higher Education Students in Alentejo, Portugal - A Cross-Sectional StudyPLOS ONE

Dear Dr. Rosário,

Thank you for submitting your manuscript to PLOS ONE. After careful consideration, we feel that it has merit but does not fully meet PLOS ONE’s publication criteria as it currently stands. Therefore, we invite you to submit a revised version of the manuscript that addresses the points raised during the review process.

It is a relevant paper about the characterization of the navigation literacy among higher students. There are several aspects to review, based on the reviewers' comments. Moreover, is there only one article reporting the prevalence of poor health literacy in Portuguese high school students (Pedro et al. in a conference abstract)? The calculated sample size of 1153 turned into a sample of 1979 participants. I think you should explain it better. Also, it is relevant to show that these participants are comparable to the entire population of Alentejo students, at least for gender and age, to be sure that the found prevalence is valid for this population. The prevalence of limited health literacy (67.5-19%) is far higher than the 44% of Pedro and the 32% of Amaral, and more in line with Santos (66%), in the University of Porto, which also points out the relevance of information sources for health education, one of the possible explanations for the results you found.

We look forward to receiving your revised manuscript.

Kind regards,

Paulo Santos, PhD

Academic Editor

PLOS ONE

Journal Requirements:

2. Please include captions for your Supporting Information files at the end of your manuscript, and update any in-text citations to match accordingly. Please see our Supporting Information guidelines for more information: http://journals.plos.org/plosone/s/supporting-information .

3. Please remove your figures from within your manuscript file, leaving only the individual TIFF/EPS image files, uploaded separately. These will be automatically included in the reviewers’ PDF.

Additional Editor Comments (if provided):

Thank you for submiting this manuscript.It is a relevant paper in the characterization of the navigation literacy among higher students.

There are several aspects to review, based on the reviewers' comments.

Moreover, is there only one article reporting the prevalence of poor health literacy in Portuguese high school students (Pedro et al. in a conference abstract)?

The calculated sample size of 1153 turned into a sample of 1979 participants. I think you should explain it better. Also, it is relevant to show that these participants are comparable to the entire population of Alentejo students, at least for gender and age, to be sure that the found prevalence is valid for this population.

The prevalence of limited health literacy (67.5-19%) is far higher than the 44% of Pedro and the 32% of Amaral, and more in line with Santos (66%), in the University of Porto, which also points out the relevance of information sources for health education, one of the possible explanations for the results you found.

Reviewers' comments:

Reviewer's Responses to Questions

**Comments to the Author**

1. Is the manuscript technically sound, and do the data support the conclusions?

Reviewer #1: Yes

Reviewer #2: Yes

2. Has the statistical analysis been performed appropriately and rigorously? 

Reviewer #1: Yes

Reviewer #2: Yes

3. Have the authors made all data underlying the findings in their manuscript fully available?

Reviewer #1: Yes

Reviewer #2: Yes

4. Is the manuscript presented in an intelligible fashion and written in standard English?

Reviewer #1: Yes

Reviewer #2: Yes

5. Review Comments to the Author

Reviewer #1: The manuscript is RELEVANT, complying with the correct scientific methodology and with PLOS ONE criteria.In my opinion, should be ACCEPT.

I have some doubts/comments that i think the authors can describe in a better way:

- in the subtitle "Study design and setting" - how was authorization obtained by the students? (via email - the authors explain it later, but not at the beginning of the article).

- In paragraph, line 151 to 157, the authors should indicate the bibliographic reference.

- In line 635, the authors should make reference to the validation of surveys for the Portuguese population.

- In the conclusion: study conclusion and data should not be compared with other studies (I suggest withdrawing). In the conclusion, only the study data and the authors' conclusions are included.

- In fact, one of the limitations of this study was mixing health sciences students with others. I congratulate the authors for naming it in the limitations. It would be interesting to carry out two separate studies in the future.

Reviewer #2: The article is well constructed and meets the objectives, however, some flaws and explanations of the process do not allow its immediate publication

Thus, I would appreciate it if the authors indicated:

1 – Why do you include the Santarém school since it is not an Alentejo school (even its NUT, the tittle define "alentejo")?

2 – How do you explain how you arrive at the sample of 952 individuals, what theory or author were you based on for this sample value?

3 – they should also explain what weaknesses exist in the HLS 16 reduced questionnaire since it is evident that the dimensions of HLS 47 are lost

4 – What happened for a potential sample of 1143 to have resulted in 1979 students?

5. Is it important to know, for example, why the question about having completed a health course previously was included?. As they wanted to evaluate undergraduate students, why is there this question?

6- Whereas 1574 students considered their health unsatisfactory They may eventually point to future studies or next steps in the research of these data, which represent a large percentage of the sample.

7- As the main reason for using an emergency service was due to unexpected problems and the aggravation of this problem, it will be useful to develop in future studies this link between lifestyle and use of emergency services.

The change from the HLS 47 questionnaire to HLS 16 jeopardizes the ability to assess the dimensions of health literacy levels, where items related to disease prevention and health promotion are limited. In this sense, it is always necessary to highlight the study by Jurgen Pelikan, who worked in depth on this HLS – EU – 47 questionnaire.

In a study of this nature it is important to confront more authors, not only those who validate. But also those who question, so this review can be enriched with a greater approach to the authors prior to M-Pohl who debated and debate this fundamental issue of assessing the level of health literacy.

https://pmc.ncbi.nlm.nih.gov/articles/PMC9659295/

Pelikan JM, Link T, Straßmayr C, Waldherr K, Alfers T, Bøggild H, Griebler R, Lopatina M, Mikšová D, Nielsen MG, Peer S, Vrdelja M; HLS19 Consortium of the WHO Action Network M-POHL. Measuring Comprehensive, General Health Literacy in the General Adult Population: The Development and Validation of the HLS19-Q12 Instrument in Seventeen Countries. Int J Environ Res Public Health. 2022 Oct 29;19(21):14129. doi: 10.3390/ijerph192114129. PMID: 36361025; PMCID: PMC9659295.

6. PLOS authors have the option to publish the peer review history of their article (what does this mean? ). If published, this will include your full peer review and any attached files.

**Do you want your identity to be public for this peer review?** For information about this choice, including consent withdrawal, please see our Privacy Policy .

Reviewer #1: No

Reviewer #2: No

---

## [Author Response · Author response to Decision Letter 1]

3 Mar 2025

Answers to Reviewer 1

Dear Reviewer 1,

We sincerely appreciate your thorough review and positive evaluation of our manuscript. Your feedback has been invaluable in refining our work, and we are grateful for your recognition of its relevance and methodological rigor. Below, we address your comments and detail the revisions made accordingly.

1. Clarification on Student Authorization in the "Study Design and Setting" Section

As suggested, we have clarified how authorization was obtained from students at the beginning of the "Study Design and Setting" section. We explicitly state that students were invited via institutional email, where they received detailed study information and a link to provide informed consent electronically before participation. The final phrase: “Following the acquisition of institutional approval, all undergraduate and integrated master's students received an email invitation via the educational institutions' communication channels. This email contained detailed information regarding the study, as well as a link to access the questionnaire. The invitation provided a comprehensive overview of the study's objectives, the participation conditions, and the significance of participation. In addition, it incorporated a link to the questionnaire. The initial section of the questionnaire encompassed all the aforementioned elements. Upon completion and understanding of the information, the students provided their consent to participate. To enhance the response rate and ensure a representative sample, follow-up reminders were sent to students, and course coordinators gave alerts during meetings.”

2. Addition of a Bibliographic Reference (Lines 151–157)

We have now included an appropriate bibliographic reference to substantiate the statement in this section, ensuring alignment with existing literature and methodological transparency.

10. Griese L, Berens EM, Nowak P, Pelikan JM, Schaeffer D. Challenges in Navigating the Health Care System: Development of an Instrument Measuring Navigation Health Literacy. IJERPH. 2020 Aug 8;17(16):5731.

36. Toolkits | M-POHL - WHO Action Network on Measuring Population and Organizational Health Literacy [Internet]. [cited 2025 Feb 14]. Available from: https://m-pohl.net/HLS19_Project

37. Pelikan J, Link T, Straßmayr C. The European Health Literacy Survey 2019 of M-POHL: a summary of its main results. European Journal of Public Health. 2021 Oct 20;31(Supplement_3):ckab164.497.

38. Pedro AR, Amaral O, Escoval A. Literacia em saúde, dos dados à ação: tradução, validação e aplicação do European Health Literacy Survey em Portugal. Revista Portuguesa de Saúde Pública. 2016 Sep;34(3):259–75.

3. Reference to the Validation of Surveys for the Portuguese Population (Line 672)

We acknowledge the importance of explicitly citing the validation process of the instruments used for the Portuguese population. We have revised the manuscript to include specific references to the psychometric validation of the HLS-EU-PT-Q16 and the HLS19-NAV in Portugal, as established in prior research (Pedro et al., 2016, 2023b; Arriaga et al., 2022).

4. Adjustment of the Conclusion

We appreciate your guidance regarding the scope of the conclusion. As recommended, we have removed comparative discussions with other studies and now focus exclusively on the study’s findings and key conclusions. Any comparative discussions are now appropriately placed in the Discussion section, ensuring that the Conclusion remains concise and focused on the study itself.

5. Acknowledgment of the Study Limitation Regarding Participant Groups

We appreciate your recognition of our acknowledgment of the limitation regarding the inclusion of both health sciences students and those from other fields. We agree that conducting separate studies for different groups of students would provide greater depth of analysis and guidance for action.

Once again, we greatly appreciate your constructive feedback and the opportunity to improve our manuscript. We believe that these revisions enhance the clarity, methodological transparency, and scientific rigor of our study.

We look forward to your further feedback and thank you for your time and expertise.

Best regards.

From the authors,

Jorge Rosário

Answers to Reviewer #2

Dear Reviewer 2,

We appreciate your constructive feedback and critical insights into our study. Your comments have allowed us to further refine the manuscript, ensuring greater clarity, methodological rigor, and alignment with the study objectives. Below, we address each of your points in detail and outline the revisions made accordingly.

1 – Justification for the Inclusion of the Santarém School

The inclusion of the Santarém school in this study is justified by its classification within NUTS II Alentejo during the 2022/2023 academic year. At that time, national statistical data classified Santarém as part of the Alentejo region. However, we acknowledge that, following the implementation of Commission Delegated Regulation (EU) 2023/674, Santarém has been reclassified under NUTS II Oeste e Vale do Tejo as of 2024 (Compete 2030). Given that our study was conducted in the 2022/2023 academic year, when Santarém was officially designated as part of NUTS II Alentejo, its inclusion remains methodologically sound and consistent with regional statistical classifications at the time of data collection.

2 – Justification for the Sample Size (952 Participants)

The sample size determination was based on standard statistical procedures used in similar studies (Pedro et al., 2023b; Sørensen et al., 2013). Specifically, we applied a 95% confidence level, assumed a population proportion of 50% (to maximize variance and ensure a conservative estimate), and set a 3% margin of error.

After applying the finite population correction, the minimum required sample size was calculated as 952 students. To account for an expected 20% non-response rate, we increased the target sample size to 1143 students. However, due to a higher-than-expected response rate, the final dataset comprised 1979 participants, exceeding the initial requirement. This increase in sample size enhanced the study’s statistical power, improved subgroup representativeness, and allowed for more precise estimates.

Revision in the manuscript: We have expanded the "Sample Size Determination" section to explicitly reference the statistical methodology used and cite relevant literature supporting our approach.

3 – Weaknesses of the HLS16 Compared to HLS47

The HLS-EU-Q16 is a validated short-form version of the HLS-EU-Q47, designed to provide a practical and efficient assessment of health literacy. However, a known limitation of using the 16-item version is that it does not capture the full range of dimensions covered by the original 47-item version. In particular, there is a reduction in the granularity of assessments related to disease prevention and health promotion (Pedro et al., 2023b).

Despite this limitation, prior research (Pedro et al., 2016; Pelikan et al., 2022) has demonstrated that HLS16 remains a valid and reliable instrument for assessing general health literacy levels, making it an appropriate choice for large-scale population studies.

Revision in the manuscript: We have expanded the "Health Literacy Assessment Tools" section to acknowledge the trade-offs associated with using HLS16 instead of HLS47, citing relevant literature (Pedro et al., 2016, 2023b; Pelikan et al., 2022). The HLS16 is also advantageous because it is shorter, i.e. it is easier to answer (making it faster) and another advantage is that more answers can be obtained (participants may not give up halfway through because it is shorter).

4 – Explanation of the Increase from 1143 to 1979 Participants

The increase in final participation numbers (1979 students vs. the estimated 1143) was primarily due to a higher-than-expected response rate. Several factors contributed to this:

• The online data collection method allowed for a broader reach and facilitated participation.

• Institutional support and engagement strategies encouraged student involvement.

• A greater willingness among students to participate in health-related studies may have contributed to the high response rate.

This larger sample size strengthened the study’s statistical power, ensuring greater representativeness and allowing for more nuanced subgroup analyses.

Revision in the manuscript: We have added a note in the "Study Sample" section explaining the discrepancy and its positive implications for data robustness.

5 – Justification for Including a Question on Prior Health-Related Education

Although the study focused on undergraduate students, the inclusion of a question about prior health-related education was based on established evidence indicating that exposure to health-related content can significantly enhance health literacy (Amaral et al., 2021). By including this variable, we aimed to explore whether previous formal training in health sciences influenced health literacy levels, thereby allowing for a more comprehensive analysis of potential determinants.

6 – Future Research on Students with Unsatisfactory Health Perceptions

We agree that the high proportion of students who perceive their health as unsatisfactory warrants further analysis. Disaggregating the data will allow for a more detailed understanding of the physical, mental and social dimensions of self-perceived health, making it possible to identify specific determinants that influence these perceptions. In addition, the development of targeted intervention strategies to improve student wellbeing is essential. We would like to emphasise that we have already started an exploratory study on this topic, which reinforces our commitment to progress in this area of research.

7 – Link Between Lifestyle and Emergency Service Utilization

Our analysis showed that unexpected health problems and worsening of existing conditions were the main factors associated with emergency care use. These findings highlight the need for future research to clarify the role of modifiable lifestyle factors in emergency care use. In particular, it is essential to investigate which emergency department visits could be prevented by behavioural change, to assess the influence of health literacy on the adoption of preventive behaviours, and to develop specific interventions to reduce overuse of these services among students. These aspects are key to guiding policies and practices that promote more efficient use of health resources.

8 – Expanding the Theoretical Framework with Alternative Perspectives

We appreciate the suggestion to engage with a broader range of authors beyond validation studies. We acknowledge the important contributions of Pelikan et al. (2022) and prior research preceding the M-POHL initiative, which have critically examined the conceptualization and measurement of health literacy.

To strengthen our theoretical foundation, we have:

• Included references to Pelikan et al. (2022) and their extensive work on HLS-EU-47 and HLS-Q12.

• Discussed alternative perspectives on health literacy assessment, emphasizing both supporting and critical viewpoints.

We sincerely appreciate your thoughtful and detailed feedback. Your comments have greatly helped to improve the clarity, methodological transparency and theoretical depth of our study. We have clarified key methodological choices, expanded the theoretical underpinnings, and suggested important directions for future research.

We believe that these revisions significantly strengthen the manuscript and look forward to your comments. We remain at your disposal. Thank you very much.

Best regards.

From the authors,

Jorge Rosário

---

## [Editor Report · Decision Letter 1]

18 Mar 2025

Navigational Health Literacy and Health Service Use Among Higher Education Students in Alentejo, Portugal - A Cross-Sectional Study

PONE-D-24-32494R1

Dear Dr. Rosário,

We’re pleased to inform you that your manuscript has been judged scientifically suitable for publication and will be formally accepted for publication once it meets all outstanding technical requirements.

Kind regards,

Paulo Santos, PhD

Academic Editor

PLOS ONE

---

## [Editor Report · Acceptance letter]

PONE-D-24-32494R1

PLOS ONE

Dear Dr. Rosário,

I'm pleased to inform you that your manuscript has been deemed suitable for publication in PLOS ONE. Congratulations! Your manuscript is now being handed over to our production team.

Kind regards,

on behalf of

Professor Paulo Alexandre Azevedo Pereira Santos

Academic Editor

PLOS ONE